# In-Line Estimation of Fat Marbling in Whole Beef Striploins (*Longissimus lumborum*) by NIR Hyperspectral Imaging. A Closer Look at the Role of Myoglobin

**DOI:** 10.3390/foods11091219

**Published:** 2022-04-22

**Authors:** Jens Petter Wold, Lars Erik Solberg, Mari Øvrum Gaarder, Mats Carlehøg, Karen Wahlstrøm Sanden, Rune Rødbotten

**Affiliations:** Nofima—Norwegian Institute for Food, Fisheries and Aquaculture Research, Muninbakken 9-13, Breivika, 9291 Tromsø, Norway; lars.erik.solberg@nofima.no (L.E.S.); mari.gaarder@nofima.no (M.Ø.G.); mats.carlehog@nofima.no (M.C.); karen.wahlstrom.sanden@nofima.no (K.W.S.); rune.rodbotten@nofima.no (R.R.)

**Keywords:** in-line, NIR spectroscopy, hyperspectral imaging, beef loin, fat marbling, myoglobin

## Abstract

Fat marbling, the amount, and distribution of intramuscular fat, is an important quality trait for beef loin (*Longissimus lumborum*) and is closely connected to sensory properties such as tenderness, juiciness, and flavor. For meat producers, it would be of value to grade and sort whole loins according to marbling on the production line. The main goal of this study was to evaluate high-speed NIR hyperspectral imaging in interaction mode (760–1047 nm) for in-line measurement of sensory assessed marbling in both intact loins and loin slices. The NIR system was calibrated based on 28 whole striploins and 412 slices. Marbling scores were assessed for all slices on a scale from 1 to 9 by a trained sensory panel. The calibrated NIR system was tested for in-line measurements on 30 loins and 60 slices at a commercial meat producer. Satisfactory accuracy for prediction of marbling was obtained by partial least squares regression for both slices and whole loins (R^2^ = 0.81 & 0.82, RMSEP = 0.95 & 0.88, respectively). The concentration of myoglobin in the meat and its state of oxygenation has a strong impact on the NIR spectra and can give deviations in the estimated marbling scores. This must be carefully considered in industrial implementation.

## 1. Introduction

Fat marbling, the amount and distribution of intramuscular fat (IMF), is an important quality trait for beef loin (*Longissimus lumborum*) and is closely connected to sensory properties such as tenderness, juiciness, and flavor [1,2,3,4]. For meat producers, it would be of value to grade and sort whole loins according to marbling on the production line. This would increase opportunities for optimal processing of this valuable meat. In the industry, entire loins can be sorted according to fat content by manual inspection. This method is quite subjective, time consuming, and not ideal in a high-volume process. A rapid and non-destructive in-line method for marbling grading of both slices and whole loins would enable efficient market driven quality sorting. In a digitized value chain, where all data can be collected and connected, such information would also be valuable for learning and value chain improvement.

There are numerous reports on how image analysis can be used to quantify fat marbling in slices or cross sections of beef and pork [5,6,7]. Over the past decade, a manually operated machine vision system has also been developed and commercialized to classify fat content/marbling at carcass level by analysis of the rib eye area between the 12th and 13th ribs of each carcass (e + v Technology GmbH, Oranienburg, Germany). A similar type of automatic machine vision would most likely work well also for the assessment of fat marbling on beef slices moving at high speed in the process, but to our knowledge, such a system is not in commercial use. Furthermore, a camera system for surface imaging would most likely be less suitable for the assessment of marbling in entire striploins since the visible fat at the surface of the muscle might not be representative of the interior.

The sensory assessed marbling score increases with the IMF content. Different studies report different relationships between sensorial marbling and IMF. The obtained correlations can also be difficult to compare since the concentration range in IMF varies between the studies. Muños et al. [7] obtained an exponential (non-linear) relation (R^2^ = 0.80) between sensorial marbling and IMF in the 1–20% range for pork. R^2^ in the range 0.76–0.79 has been reported for beef and pork [8,9,10,11]. These relationships will vary also with the skills of the assessors, whether they are consumers, trained operators, or a sensory panel. Anyhow, the measurement of IMF would give a good estimate of fat marbling.

It is well known that in-line near-infrared spectroscopy (NIRS) based on high-speed hyperspectral imaging (HSI) in combination with so called interaction measurements can be used to measure fat content in ground meat and trimmings of pork and beef [12,13]. This technology enables sampling across the entire width of a conveyor belt as well as a sampling depth of 10 to 15 mm into the meat. This technology is established in the meat industry for monitoring batch fat content. This instrumentation also allows for automatic sorting of trimmings according to estimated fat content and thereby an improved utilization of the meat raw material [14]. A limitation with NIRS and meat is that only the surface-near part of the sample is probed, which can result in highly different fat estimates of the same heterogeneous sample when measured on different sides. This was pointed out by Wold et al. [13] who reported a high prediction error (root mean square error of prediction, RMSEP) of 8.7% for single beef trimmings. This error, however, decreased with increasing trimming batch size and was at about 0.5% for batches of 120 kg.

An advantage with both whole and sliced striploins, from a measurement point of view, is that the samples are relatively homogeneous compared to beef trimmings. Loin slices are evenly sized with similar thicknesses. Whole striploins have a physiology where the fat is rather evenly distributed, meaning that NIRS interaction measurements along a loin might be representative of the internal fat content and marbling. If that is the case, the method could be useful for in-line monitoring and sorting into classes of low, medium, and high marbling scores. HSI in the NIR region has been reported for the prediction of fat marbling of beef slices [6,15], but not for whole muscles. Furthermore, most studies with HSI on meat are conducted with lab systems where the samples are either at a steady state or moving at a very low speed, e.g., 0.5 cm/s [15], while the industry requires high speed measurements and true-time analysis.

NIR interaction measurements are usually done in the 760–1100 nm region since these wavelengths have a good penetration depth in muscle. NIR spectra in this region are dominated by the absorption of molecular vibrations involving hydrogen bonds in water (O-H), fat (C-H), and protein (N-H). The water absorption peak at around 980 nm is heavily affected by sample temperature, which introduces apparent shifts [16]. Another type of shift and broadening of the water peak is related to how tightly the water is bound to proteins [17,18]. The fourth compound in beef with a strong absorption in this wavelength region is myoglobin, an iron and oxygen binding protein. Myoglobin is the main pigment giving the red color of beef. The spectroscopic properties of myoglobin in the visible region (400–760 nm) have been extensively studied and reported since they are responsible for color and color changes in meat [19,20]. A freshly cut surface of beef has a purple color given by deoxy-myoglobin. Some minutes after cutting the myoglobin is oxygenated to the form oxymyoglobin, and the color turns to fresh red. This chemical reaction, called “blooming”, might affect also the NIR spectra and should be carefully studied and understood for in-line measurements of freshly cut beef. The spectroscopic properties of myoglobin and hemoglobin in the 750–1000 nm region are reported in medical related literature [21], and the spectral signatures are being used to monitor, e.g., oxygenation in human muscle and brain [22,23].

The main goal of this study was to evaluate high-speed NIRS based on HSI in interaction mode for in-line measurement of sensory assessed marbling in both intact striploins and loin slices. The NIR system was calibrated under controlled conditions in a processing hall before it was tested for in-line measurements under realistic conditions at a commercial meat producer. The effect of myoglobin and beef blooming on the NIR spectra and prediction results were studied.

## 2. Materials and Methods

### 2.1. Materials and Experimental Design

#### 2.1.1. Calibration

A total of 28 whole beef striploins (*Longissimus lumborum*) were sampled from the production line in a Norwegian slaughterhouse. They were collected in two groups of 14 loins in each, separated in time by six months. They were selected to give a large span in fat content and marbling. Surface fat and connective tissue were trimmed away according to standard procedures, leading to samples shown in Figure 1. The loins were vacuum packed and stored at 4 °C for 2 weeks before they were shipped to our lab at Nofima for measurements.

At Nofima the loins were stored at 4 °C 3 days before measurements. The loins were unpacked, and excessive drip loss was wiped away. The following measurements were done:Every whole striploin was scanned with a NIR instrument on both sides, *lateral* and *medial*, for a total of 56 measurements. The temperature of the samples was in the range of 5–8 °C.Every whole loin was sliced into 12–15 slices of 3 cm thickness, depending on length (Figure 1). Every slice was scanned on one side with the NIR instrument, a total of 412 samples.Every slice was then photographed, on the same side as the one scanned with NIR.The photos were evaluated by a trained sensory panel that produced a marbling score for each slice.IMF was determined in two slices out of 14 loins (the second group of samples). The two slices were cut close to each of the two ends of the loin (Figure 1).

#### 2.1.2. Test in Industry

The NIR scanner was installed above a conveyor belt in a Norwegian slaughterhouse, beside the process line. This was done 8 months after collection and measurement of the last 30 freshly cut striploins from chilled carcasses were measured by the scanner. The temperature of the samples was in the range of 4–7 °C. Each loin was scanned 3 times with the *lateral* side facing the instrument. Immediately after scanning, a part of each loin (Figure 1) was cut into 5 slices of 3 cm thickness and then scanned. Two slices from each loin were shipped to Nofima for photography and determination of IMF. The photos were subsequently evaluated for marbling by the sensory panel. pH was measured in all loins to avoid DFD (dark, firm, and dry) samples with high pH. Measured values were in the range 5.40–5.76.

#### 2.1.3. Effect of Blooming

Two loin samples, low and high in IMF, were collected at the slaughterhouse, vacuum packed, and shipped to Nofima. They were stored at 4 °C one day before measurements. Two 3 cm thick slices from each loin were cut and immediately measured with the NIR system. They were then measured four more times during the next 150 min. The temperature of the samples was stable at 4 °C all the time.

### 2.2. NIR Measurements

The NIR system used was a QVision500 (TOMRA Sorting Solutions, Leuven, Belgium), an industrial hyperspectral imaging scanner designed for in-line measurement of fat in meat on conveyor belts. The instrument is based on interactance measurements in which the light is transmitted into the meat and then back scattered to the surface. The optical sampling depth in the beef is approximately 10–15 mm. The whole loins were scanned on a moving conveyor belt, and each NIR measurement took about 1 s for the entire length of the muscle. Scanning of a loin slice took about 0.1 s. The scanner was placed 30 cm above the conveyor belt so there was no physical contact between samples and the instrument. The scanner collected hyperspectral images of 15 wavelengths between 760 and 1047 nm with a spectral resolution of approximately 20 nm. The output per sample scan was an image of the loin or slice with a rather low spatial resolution. Each pixel represented a spatial area of about 7 mm × 5 mm across and along, respectively, the conveyor direction. The imaging capability was used mainly to obtain one average spectrum from each sample, but also for illustration of fat distribution within intact loins. Segmenting the sample from the conveyor belt in the images was done with a simple thresholding criterium since the spectral signature of beef was very different from the belt.

The NIR system was already calibrated for determination of fat content in beef trimmings [14] and the estimated fat values from samples in this study were used to compare results for slices of loins versus whole loins.

### 2.3. Photography of Beef Slices

Each slice of striploin was photographed in a controlled setup with a fixed distance to sample, constant camera settings (f/7.1, shutter speed 1/100 s, automatic white balance, ISO 100), and stable diffuse illumination provided by strong, external flashlights which largely dominated room lighting. A digital Canon EOS 7D was used for image acquisition.

### 2.4. Determination of IMF

IMF was determined in slices of loins using low field nuclear magnetic resonance (NMR). 3–3.5 g of homogenate was placed in a custom-made Teflon container (16 mm in diameter) with a Teflon screw top and a rubber seal for the NMR measurements. The samples were equilibrated to 40 °C for 40 min using a heat block (Dri-block heater DB-3D, Techne, Staffordshire, UK). IMF was determined using a one-shot method to estimate fat content in a 20 MHz benchtop R4-NMR spectrometer (Advanced Magnetic Resonance, Abingdon, UK) [24].

### 2.5. Sensory Evaluation of Marbling

The sensory evaluation was performed by a highly trained sensory panel at Nofima AS (Ås, Norway), consisting of 8 trained assessors trained according to ISO 8586. The sensory laboratory follows the practice of ISO 8589. A descriptive marbling analysis was performed on photographs of the loin sliced. Fat marbling was visually graded on a categorical scale from 1 (no marbling) to 9 (intense marbling). Before evaluating the marbling, the assessors were trained in a pretest (plenum session) and calibrated (individually) on selected references from grades 1, 5, and 9 (examples in Figure 2). All samples were evaluated by all assessors individually over several sessions at individual speed. Breaks were taken between the evaluation of 20–30 photos to avoid fatigue. The slices used for calibration (*n* = 412) were assessed in two equally large groups six months apart. The test set (*n* = 60) was judged eight months later. To evaluate the repeatability of the measurement, 20 samples were assessed twice by the panel. The root mean square deviation (*RMSD*) was used as a measure for repeatability, where *y_i_*_,*t*1_ and *y_i_*_,*t*2_ are sensory scores the first and second time of assessment, respectively, and *i* denotes the samples from 1 to *N*.
(1)$$RMSD=\sqrt{\frac{1}{N}\sum_{i=1}^{N}{\left(y_{i,t1}-y_{i,t2}\right)}^{2}}$$

### 2.6. Instrument Calibration

The NIR spectra were linearized using the logarithm of the inverse of the interactance spectrum (T), log10(1/T), resulting in absorption spectra. To reduce the effects of potential light scattering and small variations in distance between instrument and sample, the spectra were normalized by standard normal variate (SNV) [25]: for each spectrum, the mean value was subtracted, and the spectrum was then divided by the standard deviation of the spectrum.

In an attempt to reduce the spectral effect of myoglobin in the calibration, we did also pre-process by extended multiplicative scattering correction (EMSC) [26], where a deoxy-myoglobin like spectrum was used as a “bad” spectrum (the difference spectrum shown in Figure 4b).

Partial least squares regression (PLSR) was used to make calibrations between NIR spectra and marbling scores. Cross validation was applied to determine the optimal number of PLS factors and to evaluate the model’s predictive ability. Replicate measurements of the same samples were left out in the same cross validation segment. Beef slices from the same striploins were also grouped in the same cross validation segments to avoid overfitting. The prediction error was estimated by the root mean square error of cross validation (*RMSECV*) where *ŷ_i_* is the predicted value from the cross validation, *y_i_* is the reference value and *i* denotes the samples from 1 to *N*.
(2)$$RMSECV=\sqrt{\frac{1}{N}\sum_{i=1}^{N}{\left(y_{i}-\hat{y_{i}}\right)}^{2}}$$

For calibrations for whole loins, the response value used per loin was the average marbling score of all slices from that loin and full cross validation was used.

The calibrations were implemented in the NIR system to produce marbling values in true time. The calibrations were tested on new whole striploins and slices of loins in the commercial slaughterhouse.

The software The Unscrambler version 9.8 (CAMO Analytics AS, Oslo, Norway) was used for calibration and analysis of the data. Image processing and spectral preprocessing were carried out in MATLAB version 7.10 (MathWorks Inc., Natic, MA, USA).

## 3. Results and Discussion

### 3.1. Sensory Assessment of Marbling

The 412 calibration samples spanned the full range of marbling scores from 1 to 9. The scores used in this study were the average score per sample for the eight judges. The 20 samples that were judged twice by the panel indicated an RMSD of 0.43, which can be regarded as a high repeatability. Another interesting measure is the standard deviation of the judge’s scores per sample since it says more about expected variation from person to person. The mean standard deviation was 0.85.

The marbling scores were closely related to the concentration of IMF (R^2^ = 0.86), as measured in 88 loin slices, illustrated in Figure 3. This relationship was clearly non-linear and very similar to results obtained on slices of pork ham [6]. Others report linear relationships, but these have often been obtained on meat with considerably higher fat levels [4].

The reason for non-linearity is not clear. Muños et al [6] suggested that samples with high marbling scores contain more small deposits of fat between muscle fibers that are difficult to discern for the human eye. Another explanation could be that the adipose tissue in very marbled samples has a higher concentration of fat. We checked this by measuring the concentration of fat in adipose tissue from four samples but found no support for this hypothesis. The psychological literature refers to the Weber-Fechner law which states that linear increments in sensation are proportional to the logarithm of the stimulus intensity [27]. This relationship was initially intended for perceptual faculties such as loudness or brightness, but does also apply to more abstract parameters [28]. This may explain why the relationship between IMF and marbling is non-linear.

There was a large variation in marbling between the loins, and there was also quite some variation in marbling within some of the loins. Figure 8 shows examples of marbling distribution in three loins. The biggest internal variations were found in rather fat loins.

### 3.2. NIR Spectra and Fat Estimates

Typical NIR spectra from loin slices are shown in Figure 4a. The broad peak at around 980 nm is absorption by water. Fat absorbs at around 930 nm but there was no clearly visible fat peak since the highest concentrations were just above 20%. It is clear that the apparent offset of the spectra correlated positively with marbling (R = 0.74 at 860 nm). The more fat, the higher absorption. This was not expected. Wold et al. [14] reported the opposite for beef trimmings measured with the exact same NIR system: Samples with high fat content had lower absorption than lean beef. Adipose tissue does in general have lower absorption than beef muscle. In beef trimmings, there are large shares of pure adipose tissue, while in the loin the fat is marbled into the meat. It is well known that older animals have higher concentrations of myoglobin in the muscle [29]. Meat from older animals does also tend to contain more fat. A close inspection of the spectra indicated that differences in myoglobin concentration was the main contributor to the large variations in offset. Dark red samples had an overall higher absorption than light red beef. Even on beef with a lot of marbling, where adipose tissue made up much of the surface, this was the case because the red muscle fibers were darker. This can be observed in Figure 2 where the most marbled slice had darker muscle fibers compared to the leaner samples. Figure 4b shows absorption spectra from two very lean samples where one is dark red and one is light red. There was a clear offset between the two, and when the lower spectrum is subtracted from the higher we get a difference spectrum which is very similar to the absorption spectrum of myoglobin [21,22].

In the beef spectra, myoglobin is responsible for the steep absorption towards 760 nm, and a relatively sharp shoulder around 920 nm. The variation in myoglobin seemed to represent as much as 95% of the variation in the calibration set (based on principal component analysis, data not shown). It is very important to be aware of the huge impact of myoglobin on these spectra. First, the shoulder at 920 nm is close to the absorption peak for fat at 930 nm and might affect IMF and marbling estimates unless this is thoroughly handled in the calibration. Second, the absorption spectrum of myoglobin is not stable over time since it changes with the degree of oxygenation. Figure 4c shows how the NIR spectrum from a freshly cut loin surface changes during the first 150 min of exposure to air.

Deoxy-myoglobin gradually turns to oxy-myoglobin. This is seen by a systematic lower absorption at 760 nm and a higher absorption for wavelengths longer than 800 nm. We have included a spectrum measured after 3 days when most of the myoglobin in the upper 10 mm layer is in the oxy state to illustrate the huge spectral differences. The changes observed here correspond well with published spectra of deoxy- and oxy-myoglobin [21,22]. It should be noted that the surface color of the sample stabilized after about 20 min of exposure to air, while the blooming effect continued in depth of the sample as illustrated by others [30]. Within NIR spectroscopy we normally model on rather subtle spectral differences, so it is obvious that the changes induced by myoglobin and oxygenation will most likely affect predictions of fat/marbling unless they are accounted for in the calibration or avoided in the industrial process by measuring all samples at approximately the same time after cutting and trimming.

The fat calibration that was already implemented in the NIR scanner was used to estimate the fat content in all measured samples in the calibration set. This was used to compare fat estimates for whole loins versus slices. Figure 5 shows that there was a very close linear relationship (R^2^ = 0.98) between estimated fat in the whole loins with the average estimate for all slices from the same loin. This indicates that the NIR spectra measured along the upper layer of intact loins were highly representative of the internal IMF. Figure 5 represents NIR measurements done on the lateral side of the loins. Measurements on the medial side resulted in a slightly lower correlation (R^2^ = 0.96) maybe because remnants of surface fat affected the measurements.

### 3.3. Modelling of Marbling Based on NIR

Table 1 summarizes the calibration and testing of different regression models for marbling. Models were made for both loin slices and entire loins. Calibration performance was quite the same for absorption spectra and SNV corrected spectra for the slices, while SNV performed slightly better for whole loins. EMSC correction with the aim of removing the effect of myoglobin did not work well. Since the concentration of myoglobin correlated quite well with marbling and fat, as described above, the removal of this spectral information was not successful. The choice of the correction spectrum for myoglobin might be critical since it has a shoulder close to the fat peak.

SNV would also remove the offset variation connected to myoglobin, but the information about fat was still maintained in the spectra. SNV data required one PLS factor less than non-normalized data, which is quite usual. The SNV based calibrations were implemented in the NIR instrument and used for true time measurements at the meat producer. When a calibration is made at one location and the NIR instrument is moved to another location, it is not unusual that a slope and/or an offset is encountered in the new prediction values. This was, to some extent, the case also now. These offsets are given in Table 1. For practical use in the industry, such offsets would be subtracted as a very simple way of calibration transfer. Before the RMSEP values in Table 1 were calculated, the offsets were therefore subtracted.

The results for the prediction of marbling in new samples in an actual industrial line were quite similar to those of the calibration, in particular for SNV corrected spectral data. Figure 6 shows predicted versus sensory assessed marbling for single slices and whole loins. R^2^ values above 0.80 are rather high when modeling is based on sensory data. A prediction error (RMSEP) of about 0.90 is on par with the average standard deviation for the judges (0.85). Having in mind that the sensory evaluation was performed in three sessions (calibration set 1, calibration set 2, test set) more than six months apart, we can assume that the scoring procedure had a high degree of repeatability. The correlation between estimated marbling scores from whole loins versus the average of the slices was again high (R^2^ = 0.92), verifying that NIR measurements on intact loins were representative of the internal marbling. A lower correlation than what was obtained for fat estimates for the calibration set (Figure 3) might be due to the fact that only five slices per loin were measured for the test set, versus 15 in the calibration set (see Figure 1). The average of these five would be less representative for the whole loin and a lower correlation can be expected.

In this work, we used linear regression for calibration. Since there was a clear non-linear relationship between IMF and marbling score, a non-linear modeling approach could be more appropriate. For instance, a linear calibration could be used to estimate the IMF based on NIR, and then the marbling could be estimated based on the relationship with IMF (Figure 3).

During the test measurements, we observed some deviations from the expected. Two very lean loins got high marbling scores in the range 3–4 based on the NIR calibrations, while another lean loin got a low value of about 1, as expected (samples singled out in Figure 6). The loins with overestimated scores had dark red color, while the ones with low estimated scores had a very light red color. These deviations in prediction can most likely be attributed to myoglobin. A closer look at the predicted values in Figure 6 revealed that dark samples tended to get slightly higher scores than lighter red samples. When we started this work, we were not aware of the influence myoglobin would have on the spectra. Ideally one should collect calibration samples where the concentrations of fat and myoglobin are very weakly correlated so that calibrations of fat/marbling will not depend on variations in myoglobin.

We also observed that a dark loin that obtained a marbling score of 4, when scanned 15 min later got a score of 3. The spectral effect of blooming, as shown in Figure 4c, affected the estimated marbling scores. Figure 7 shows how estimated marbling scores varied with time of air exposure. Note that the estimated score dropped by about 1 unit during the first 20 min. Ideally, a calibration should be invariant to the effect of blooming. This can most likely be done by including this variation in the calibration set. The timing of NIR measurements after cutting the beef was not strictly controlled during calibration and testing in this work. A strict timing regime would reduce the effects of blooming and potentially also improve performance.

To our knowledge, the effect of myoglobin on fat calibrations for meat by NIRS is hardly discussed in the literature. This might be due to (1) unawareness of its role or (2) because myoglobin in other meat systems and with different NIR technology will have less effect than in the present study. Myoglobin absorbs strongly in the 760–1047 nm region while the absorption of deoxy and oxy states is much more limited at wavelengths longer than 1200 nm [31]. The work of Kuenstner and Norris [31] was done on human hemoglobin but is still very relevant since hemoglobin has very similar spectral properties to myoglobin.

Measurements in our study were done in interactance mode to probe as deeply as possible. The radiation traveling distance in the meat is then much longer than for reflection measurements. For dark beef samples with high levels of myoglobin, there might be a chance that radiation at some wavelengths are almost completely absorbed and that spectra become distorted. This kind of data will be difficult to fit into a model and therefore important to detect by some kind of outlier detection. For a NIR interaction system used on this kind of application, there will be a tradeoff between probing as deep as possible and still obtaining undistorted spectra with a high signal to noise ratio. The distance of interaction can be adjusted to optimize this tradeoff, as pointed out by Wold et al. [32]. NIR spectroscopy in reflection mode will also be sensitive to blooming, and the complete transition from deoxy-myoglobin to oxy-myoglobin will most likely be much more rapid since mainly the surface of the meat is probed. As indicated in Figure 4c this transition took as much as three days with interaction measurements since they probe deeper and can therefore measure the gradual diffusion of blooming in depth.

Trained workers can probably grade whole loins into marbling classes with quite a good accuracy. A well-working in-line classification instrument would still be an advantage since speed would be no issue and the measurements would be objective day after day. The ability to non-destructively score the fat marbling on whole loins gives the processor a chance to consider the optimal use of each and every loin early in the process. The hyperspectral imaging system used in this study can give more information than just an average score per loin. Figure 8 shows that estimated fat marbling can be imaged on every loin. The image patterns correspond to some degree with the actual marbling measured on the slices. The marbling pattern might serve as a guide for portioning the loins into parts of different quality, dedicated to different customers.

## 4. Conclusions

In-line estimation of fat marbling in whole beef loins by NIR hyperspectral imaging can be achieved with acceptable accuracy. This is possible because the IMF content in the upper layer of the lateral side of the loin correlates well with the internal fat content in the loin. Myoglobin and its state of oxygenation has a strong impact on the NIR spectra and must be carefully considered at implementation. A robust calibration that can handle the variations of myoglobin is one solution and an interesting topic for future work. This work also emphasizes the importance of developing and evaluating NIR applications under realistic conditions to discover critical factors that affect the performance of the instrumentation.

## Figures and Tables

**Figure 1 foods-11-01219-f001:**
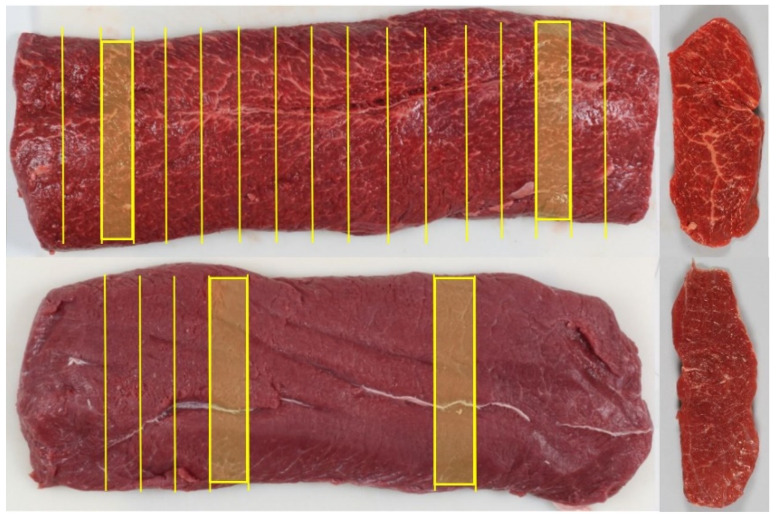
Two samples of whole striploins and belonging slices (right). Cutting pattern on upper loin was used for the calibration data set. Cutting pattern on lower loin was used for the test set. All cut slices were measured with NIR. Fat content was determined in shaded slices. Sensory scoring of marbling was done on all slices in calibration set and for the shaded slices in test set.

**Figure 2 foods-11-01219-f002:**
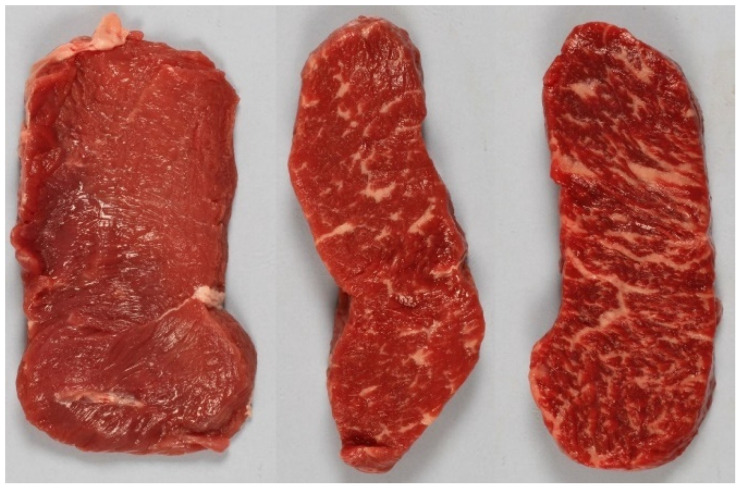
Reference slices for fat marbling. Scores 1 (**left**), 5 (**middle**) and 9 (**right**).

**Figure 3 foods-11-01219-f003:**
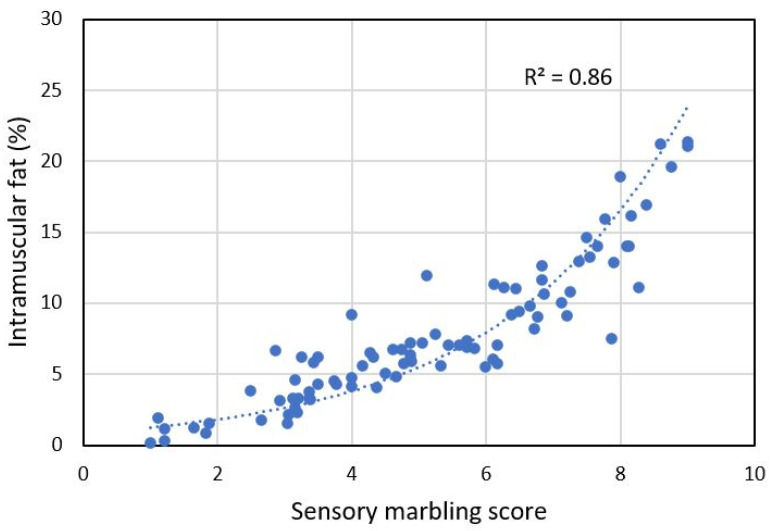
Relation between sensorial fat marbling scores and IMF for sliced loin steaks.

**Figure 4 foods-11-01219-f004:**
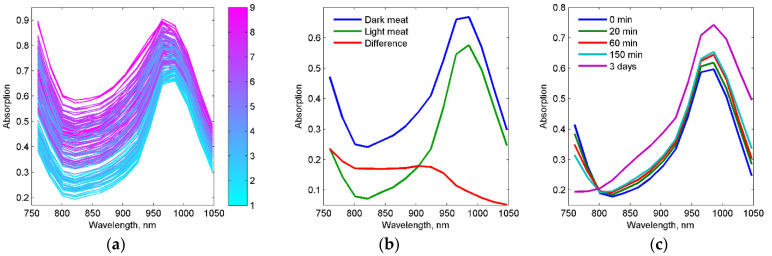
(**a**) NIR calibration spectra from loin slices. Colorbar indicates marbling scores for the slices. (**b**) NIR spectra from light and dark beef, including difference spectrum. (**c**) Spectral effects of blooming—oxygenation of myoglobin.

**Figure 5 foods-11-01219-f005:**
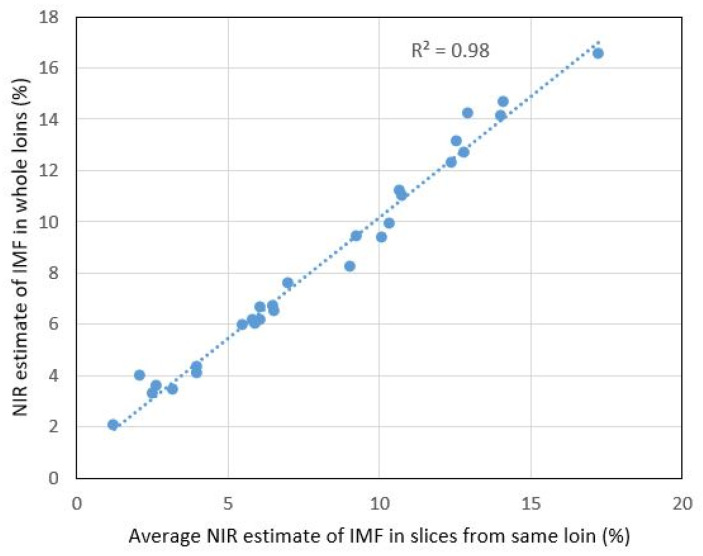
Relation between NIR fat estimates for intact loins and average fat estimates for the loin slices.

**Figure 6 foods-11-01219-f006:**
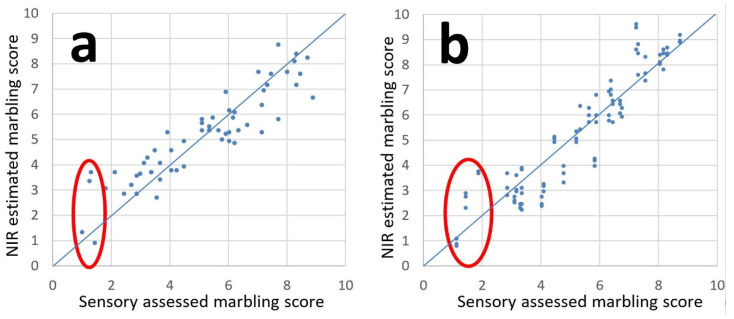
Predicted versus sensory assessed marbling scores in (**a**) single slices and (**b**) entire loins.

**Figure 7 foods-11-01219-f007:**
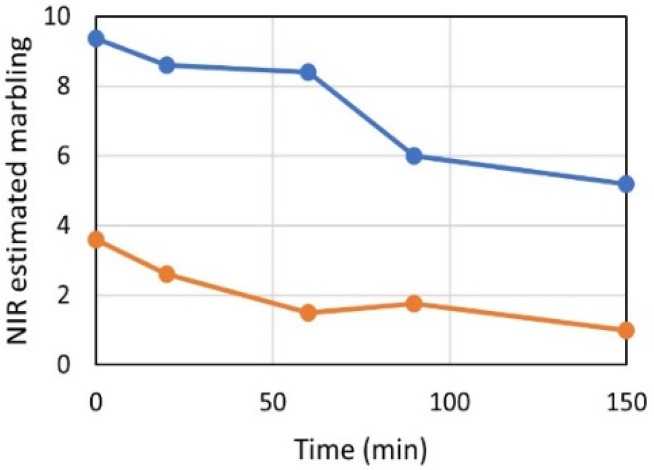
NIR estimated marbling scores over time for fat (blue) and lean (orange) loin slice.

**Figure 8 foods-11-01219-f008:**
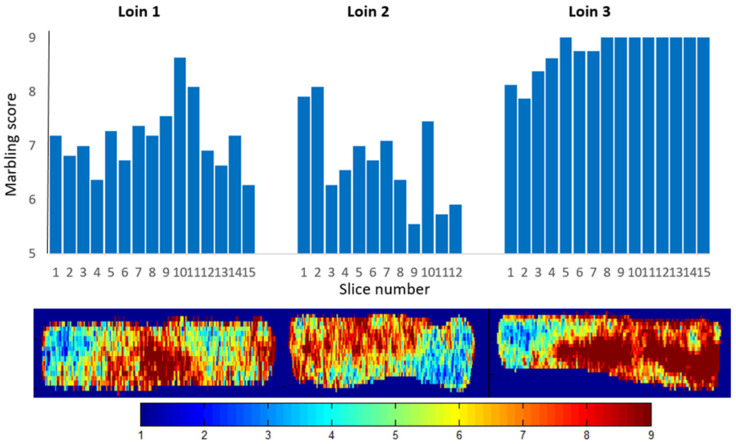
Images of estimated fat marbling in entire loins aligned with sensory assessed marbling in slices from the same loins. Colorbar indicates predicted marbling scores at pixel level.

**Table 1 foods-11-01219-t001:** Results of calibration and testing in industry.

Loin Slices	Calibration (*n* = 412)			Test In-Line (*n* = 60)		
	#LV ^a^	R^2^	RMSECV ^b^	R^2^	RMSEP ^b^	Offset
Abs	5	0.81	1.0	0.82	0.89	0.7
SNV	4	0.81	1.0	0.81	0.95	1.0
EMSC	2	0.26	2.1	0.15	2.20	0.1
**Whole Loins**	**Calibration (*n* = 28)**			**Test In-Line (*n* = 30)**		
Abs	5	0.82	0.98	0.76	1.14	0.7
SNV	4	0.87	0.82	0.82	0.88	0
EMSC	2	0.24	2.1	0.15	1.93	0.1

^a^ Number of latent variables used in model. ^b^ Root mean square error of cross validation/prediction.

## Data Availability

Data is contained within the article.

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
