# Peer review of "In-Line Estimation of Fat Marbling in Whole Beef Striploins (*Longissimus lumborum*) by NIR Hyperspectral Imaging. A Closer Look at the Role of Myoglobin"

_foods, 2022, doi:10.3390/foods11091219_

Round 1

Reviewer 1 Report

The concept of quality in relation to beef is multidimensional and includes several groups of factors, the methods of its evaluation must be adapter to their characteristics. Currently, objectify and shorten the time of analysis based on non-invasive and non-destructive research methods are the aim of different research. One of these methods is near-infrared spectroscopy due to the flexibility and high rate of measurement of various types of food products. While, near infrared spectroscopy - NIR can provide on-line functionality of the measurement due to its rate and ease of use, in addition it gives the opportunity to evaluate many quality parameters during one measurement. However, using the NIR method requires proper calibration and multiple tests.  In this paper, the authors attempt to use high-speed NIR hyperspectral imaging for in-line measurement of sensory assessed marbling in both intact loins and loin slices. This is in line with recent trends in terms of the testing methods used, as well as the verification of meat quality, especially beef, which is expensive and time-consuming to produce. Verification of beef quality at an earlier stage of production can significantly reduce financial losses. The whole experiment was prepared and performed properly and met all the requirements of scientific experiment. The authors obtained accuracy for prediction of marbling was obtained by partial least squares regression for both slices and whole 19 loins (R2 = 0.81 & 0.82, RMSEP = 0.95 & 0.88, respectively). The authors postulate that the use of high-speed NIR hyperspectral imaging for in-line of fat marbling in whole beef loins is possible. I support this claiming and I consider this research to be of great value, especially taking into account the need to develop rapid and non-destructive methods of beef quality assessment, allowing the analysis of multiple parameters simultaneously. I recommend minor revision. All my remarks are listed below:

Line 144 At what intervals were measurements taken during blooming were color changes still significant after 60 min?

Line 408 please the header should be above the table 1

Line 476 Figure 7 please change the line color or description because this line is not red

In addition,whether determination of myoglobin and its forms (quantitatively) could improve the results (fitting the prediction model)?

Author Response

Thank you very much for the reviewers time and the favourable reviews. Below are comments to the referees.

Reviewer 1:

Line 144 At what intervals were measurements taken during blooming were color changes still significant after 60 min?

The time intervals were 20, 60, 90, 120 and 150 minutes and are now given in the text. The colour changes were stable after about 20 minutes (also included in the result part).

Line 408 please the header should be above the table 1

Header has been moved.

Line 476 Figure 7 please change the line color or description because this line is not red

Description has been changed.

In addition, whether determination of myoglobin and its forms (quantitatively) could improve the results (fitting the prediction model)?

Yes, it might be a point to quantify the myoglobin and its two forms to obtain better models. At least to understand the models and the variation in full.

Reviewer 2 Report

The study investigated the feasibility of in-line detection of beef marbling using NIR hyperspectral imaging. The results indicated that in-line estimation of fat marbling in whole beef loins by NIR hyperspectral imaging could be achieved with acceptable accuracy. Overall, the study with an informative data and a profound discussion has great practical value. However, the English and technical presentations of the manuscript should be further improved. Moreover, the description of some methods is not detailed enough. Therefore, the manuscript needs to be revised carefully.

  1. The detection of beef marbling usingNIR hyperspectral imaging is common. What is the innovation in this study?
  2. English and technical presentationsshould be further improved.
  3. The introductionwas incoherent and was not well organized. Authors should go through the introduction carefully and conceive a better writing structure, which make it easy for readers to understand.
  4. In the manuscript, the temperature ofsample detection was inconsistent. For instance, line 119, line 132 and line 145. Why?
  5. Line 120, the whole loin was sliced into 13-15 slices. However, in Fig. 8, the slice number of loin 2 was sliced into12 slices, which should be unified in the manuscript.
  6. Line 120, what's the thickness of beef slices?
  7. Line 177-178. The wavelengths from760 to 1047 nm were selected. Why?
  8. In section 2.5, what method wasused in cross validation?
  9. Myoglobinwas mentioned in the tittle. However, a profound discussion about that was missing. In addition, the absorption peak of myoglobin is at 480 nm and the absorption changes caused by protein denaturation are mainly at 910 nm. Therefore, the discussion (Line 336-338) was not enough to explain that the spectral changes were caused by the oxygenation of myoglobin. A profound discussion should be provided.

Author Response

  1. The detection of beef marbling usingNIR hyperspectral imaging is common. What is the innovation in this study?

HSI in the NIR region has absolutely been reported for prediction of fat marbling of beef slices, but not for whole muscles as we do in this study. Furthermore, most studies with HSI on meat and marbling are conducted with lab systems where the samples are either at steady state or moving at very low speed, while the industry requires high speed measurements and true-time analysis. In this study we use HSI at high speed under industrial conditions to estimate marling in both slices of beef loins and whole striploins. This is novel.

  1. English and technical presentations should be further improved.

We have overseen the English and hope it is acceptable.

  1. The introduction was incoherent and was not well organized. Authors should go through the introduction carefully and conceive a better writing structure, which make it easy for readers to understand.

We agree that the introduction could be perceived as incoherent. It has now been simplified and made more to the point. It should now be easier to follow.

  1. In the manuscript, the temperature ofsample detection was inconsistent. For instance, line 119, line 132 and line 145. Why?

5-8C was used to mimic industrial conditions, 4-7C was the real industrial conditions, so very close. The temperature range is not extremely critical and the most important is that it spans a realistic range that will be encountered in the industry. 4C for the blooming experiment was chosen as a typical temperature for storage of meat.

  1. Line 120, the whole loin was sliced into 13-15 slices. However, in Fig. 8, the slice number of loin 2 was sliced into12 slices, which should be unified in the manuscript.

This is corrected to 12-15, thank you.

  1. Line 120, what's the thickness of beef slices?

Thickness was approximately 3 cm. This is now included in the text.

  1. Line 177-178. The wavelengths from760 to 1047 nm were selected. Why?

This range was selected since penetration depth in meat is good and better than NIR at wavelengths longer than 1100 nm.

  1. In section 2.5, what method was used in cross validation?

Cross validation for beef slices: Cross validation was applied to determine the optimal number of PLS factors and to evaluate the model’s predictive ability. Replicate measurements of the same samples were left out in the same cross validation segment. Beef slices from the same striploins were also grouped in the same cross validation segments to avoid overfitting.

Cross validation for entire striploins: Full cross validation. This is stated in the text.

  1. Myoglobin was mentioned in the tittle. However, a profound discussion about that was missing. In addition, the absorption peak of myoglobin is at 480 nm and the absorption changes caused by protein denaturation are mainly at 910 nm. Therefore, the discussion (Line 336-338) was not enough to explain that the spectral changes were caused by the oxygenation of myoglobin. A profound discussion should be provided.

I is not correct that myoglobin has an absorption peak at 480 nm. The visible region is anyway not of great interest for this study, since we are measuring in the 760-1047 nm range. Myoglobin has strong absorption in this region, both the oxy and deoxy forms. We do not agree that the discussion concerning myoglobin is insufficient. It is not over just two lines as the referee indicates but over several paragraphs, lines 327-370 and 452-498. So we prefer to leave it almost as it was.